# Unravelling the Role of Institutions in Market-Based Instruments: A Systematic Review on Forest Carbon Mechanisms

Xinran Shen [1,*], Paola Gatto [2] and Francesco Pagliacci [2]

1   Ph.D. LERH Program, Department TESAF, Land, Environment, Agriculture and Forestry, University of Padova, 35020 Legnaro, Italy
2   Department TESAF, Land, Environment, Agriculture and Forestry, University of Padova, 35020 Legnaro, Italy
*   Correspondence: xinran.shen@phd.unipd.it

**Abstract:** Forest ecosystems provide various services that are crucial to human beings, in which carbon sequestration and storage is one of them with the most market potential and is usually governed by market-based instruments (MBIs). MBIs do not operate alone but in the hybrid governance arrangements. While the importance of public institutions has been identified, there is still a need to examine the specific role of public institutions in the market-oriented mechanism. Our work seeks answers to this question. This meta-study presents an up-to-date picture of MBIs targeted at forest carbon, in which 88 mechanisms are synthesized in a quantitative database. We analyze and discuss policy design features of these mechanisms and group them into nine types of MBIs. We find that many instruments coexist and/or interact with other instruments. In light of these results, we introduce the concept of policy mix and argue that the interplay among policy instruments can be complementary or interdependent. Using cluster analysis to identify underlying patterns, we reconfirm previous findings that there are distinct differences between public and private PES schemes, but also recognize a new cluster and label it as a 'legally binding mechanism'. We discover that the role of public institutions is pronounced in the forest carbon mechanisms, and they can be the buyer, seller, regulator, coordinator, intermediary, and facilitator. Besides, public institutions tend to play an increasing role in the future climate policy arena. We believe that public institutions should stand out and create enabling conditions for private governance and finance.

**Keywords:** policy instrument; forest; incentive; governance; PES; cap-and-trade; systematic literature review

## 1. Introduction

During the previous decade, market-based instruments (MBIs) have been mainstreamed in environmental and forest governance [1]. Compared with so-called 'command-and-control' approach, MBIs govern public goods by markets and market values [2]. This increasing trend can be explained by the fact that MBIs can improve economic efficiency while providing forest collective values and conserving forest multifunctionality [3–5]. Forests provide a wide range of ecosystem services (ES), including fuel and fiber, flood control, water supplies, clean air, etc. [6]. Among these, carbon sequestration and storage (also referred as climate regulation) is cited as one of the ESs with the most market potential [7]. Forests make a great contribution to mitigating carbon emissions by acting as a carbon reservoir and a tool to sequestering carbon through photosynthesis. The market for forest carbon is structured, well-developed, and institutional, and it is closely linked to international and national agreements and polices, such as the Kyoto Protocol (KP), Paris Agreement (PA), and European Green Deal. Policy contexts can shape the way single political decisions take place. For example, the Clean Development Mechanism (CDM), under the KP, finances afforestation and reforestation projects in developing countries [8]. Reducing Emissions from Deforestation and Forest Degradation (REDD+), under the United

Nations Framework Convention on Climate Change (UNFCCC), links climate change and international forest policy through incorporating forest carbon projects in the voluntary system [9].

The term 'MBI' is sometimes used interchangeably with 'economic instruments', 'incentive-based instruments', or 'payment for ecosystem services (PES)'. For the sake of clarity, MBI is defined in this study as a broader range of financing instruments, following the definition and classification provided by SINCERE (Spurring INnovations for Forest ECosystem SERvices in Europe (https://sincereforests.eu/accessed on 10 December 2021)) project, funded through the European Commission's Horizon 2020 program [10]. Besides, the notion of 'PES' follows the classical definition by Wunder [11]. In that project, Bottaro et al. divided MBIs into three types, including quantity-based instruments (i.e., cap-and-trade schemes, mitigation banking, offset scheme), price-based instruments (i.e., subsidies, tax exemption, competitive tenders/auctions), and market-friction reducing instruments (i.e., PES and PES-like schemes, eco-certification, corporate social responsibility, public-private contracts) [10]. As MBI literature flourishes and new case studies are continuously being added, a global-comparative study presenting an up-to-date picture is needed [12].

It has been well-documented that real-world conservation is more complicated than what individual policy instrument can handle. Possible policy challenges could be heterogeneous and multiple objectives of ES provision, mix of values of forest ecosystems, (multiple) market failures, multi-level governance (require appropriate instruments at different levels), etc. [13]. Instead of ideal MBIs, policy mixes (the use of multiple policy instruments) are more likely to be applied to solve those challenges [14]. For example, Barton et al. argued that the well-known PES program in Costa Rica is a policy mix rather than a single economic instrument [15]. However, previous studies are usually limited to individual policy instruments and completely disregard potential interactions between instruments [16–18].

Gómez-Baggethun and Muradian [4] argued that the market cannot operate without the active intervention from public institutions; therefore, the notion of MBIs contains both market and command elements. Previous findings also confirmed that public institutions play overarching roles, and MBIs are not all about moving from public polices to market allocations [19–21]. Public supports are usually provided through the adoption and enforcement of regulations, establishment of market infrastructure, or clarification of land tenure and property rights [22]. The role of public institutions is different for different types of MBIs as well as for the mechanisms it encompasses [23]. There have been many studies actively promoting the 'market' element of MBIs while completely ignoring the 'command' element [3,24]. Bearing this in mind, an in-depth analysis about the role of public institutions in MBIs is necessary.

The objective of this article is threefold. First, this study is aimed at mapping global and up-to-date empirical patterns of MBIs used in carbon sequestration and storage, with a strong focus on forest ecosystems. Second, this study is aimed at deepening the understanding of MBIs by quantitatively identifying and discussing their policy design features. Third, this study is aimed at unravelling the role of institutions in the governance of MBIs targeted at forest carbon. By adopting a worldwide systematic literature review, a sample of 88 mechanisms are retrieved from the Scopus database. Compared with meta-studies like Grima et al. [25] and Berthet et al. [16], this study includes more cases and the most up-to-date information.

This article is structured as follows. In Section 2, the processes of systematic literature review and methodology applied to mechanisms are explained in detail. Section 3 reports research outcomes regarding statistical analysis of retrieved literature. Also, the typology of different policy instruments and actors are presented in the Section 3. Potential insights of this analysis, and their implications for future research and policies are discussed in the Section 4. Finally, conclusions are given in the Section 5.

## 2. Materials and Methods

### 2.1. Data Gathering

With diverse and fragmented typologies of MBIs, a systematic literature review is applied to compile, appraise, and evaluate research on forest carbon (See Figure 1). The review followed the guidelines by previous studies and was conducted according to the PRISMA approach [26–28]. Two-round literature searches were conducted on Scopus to identify relevant peer-reviewed papers. Key search terms are shown in the Table S1. The results from the two searches were combined and repetitions were excluded. We limited the search to a period ranging from 2006 to 2021. This paper chose 2006 as a starting year as the ES concept was first made public by MEA in 2005 [6]. The literature search was limited to English language results. There was no limitation to the geographical location of the studies. ES category was based on The Common International Classification of Ecosystem Services (CICES) V5.1, which is one of the most common classifications of ES [29]. It is worth mentioning that particular attention has been given to mechanisms targeted at forest carbon, but some mechanisms dealing with bundled ES (including carbon-related ES) were also involved due to the close relationship among different types of ES.

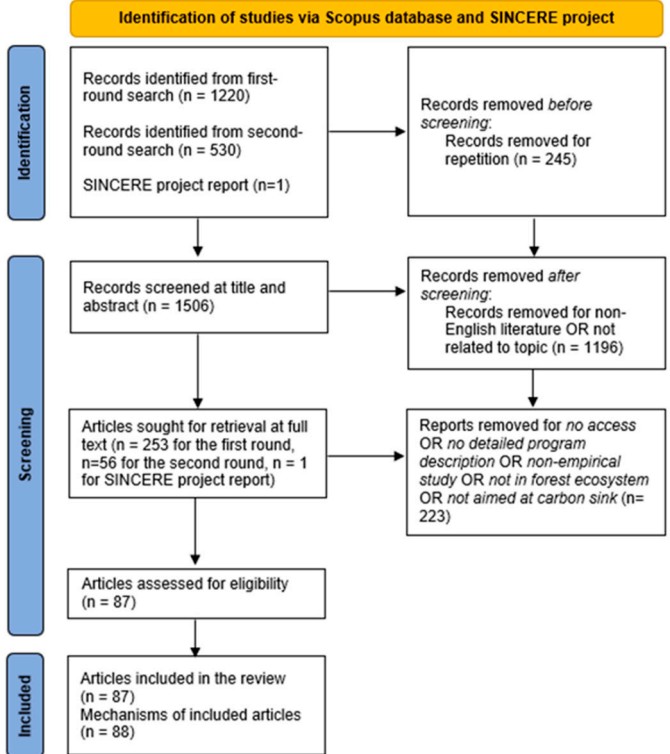

**Figure 1.** Diagram of systematic literature review (adapted from Page et al. [30] and Maier et al. [28]).

The literature was first identified through keyword search, then records were screened by the title and abstract, and if relevant to the topic, they were screened at the level of the full text. Although hundreds of studies reported MBIs for forest carbon, most literature did not provide in-depth information for the meta-analysis. During the screening process, articles or studies were selected based on the following criteria:

- empirical work from peer-reviewed literature that focuses on:
- market-based instruments,
- aimed at carbon sequestration and storage, and
- having a strong focus on forest ecosystems.

## 2.2. Data Analysis

We extracted both qualitative and quantitative data from previously published literature, gathered demographic information for each mechanism, and coded them with the common terminologies (see Supplementary Material File S2). After creating profiles for each market-based mechanism, a database was established, and qualitative comparative analysis was used to synthesize empirical findings. Through the establishment of the database, a larger population could be generalized to, statistical power was enhanced, and how different contexts shaped the relationship between mechanism and outcome could be explored [31]. The information from literature search was summarized and documented at the article level and the mechanism level, following the guideline by Martin-Ortega et al. [32]. The article level refers to those variables across articles (e.g., author, publication year, journal type, reference). The mechanism level corresponds to those variables across mechanisms (e.g., case study location, instrument types, funding sources). This study also intended to identify the role of public institutions in MBIs and assess the degrees of public institutions involvement. The qualitative information about the role of public institutions was identified and analyzed, through multiple correspondence analysis (MCA) and cluster analysis. MCA is an extension version of principal component analysis (PCA) for categorical variables, which is aimed at representing multivariate datasets as a smaller set of variables (summary indices) to reduce data dimensions and provide simpler views in graphs [33]. Cluster analysis was performed to find similar groups of subjects using hierarchical clustering method (distance measure: Euclidean distance). With the hierarchical clustering procedure, entities were grouped together starting with one entity in each group and continuing until reached the number of clusters that was specified, or all the entities were in one group [34]. Both analyses were performed in R using the package FactoMineR to explore underlying patterns in mechanisms and identify trends for the application of MBIs [35,36].

## 3. Results

### 3.1. Descriptive Analysis Results

By running a systematic review, 87 articles published between 2006 and 2021 were finally selected and analyzed, and 88 market-based mechanisms were identified from these. Some mechanisms only target single ES, while other mechanisms target bundled Ess. Table 1 shows that the most frequently mentioned Ess in the review are climate regulation (45%); lifecycle maintenance, habitat, and gene pool protection (22%); and water quality regulation (10%). The less frequently mentioned Ess include water provision (6%), timber production (5%), and cultural ES (4%). The cultural Ess targeted in mechanisms are aesthetic/heritage, recreation and eco-tourism, and educational ES.

MBIs can adopt different project activities to increase the quantity or improve the quality of target ES. There can be more than one project activity implemented in a mechanism. The main ones are afforestation/reforestation (34%), improved forest management (24%), reducing emission from deforestation and forest degradation (18%), and agroforestry (17%), which account for over 90% of project activities. There are four mechanisms regarding avoiding conversion of grasslands and shrublands to crop production and three mechanisms concerning restoration of wetland/peatland. Only one project belongs to the biomass energy project category and this project is focused on improved charcoal production for sustainable biomass use. The temporal distribution of establishment of mechanisms is shown in the Figure S1. An increasing trend can be observed between 2001 and 2008. Following that, the number of new mechanisms kept decreasing.

**Table 1.** Ecosystem service types, counts, and their proportions out of the total counts (proportion represents the mechanisms that have this type of ES account for the total mechanisms).

| Ecosystem Service Type * | Count | Proportion |
|---|---|---|
| Climate regulation | 86 | 45% |
| Lifecycle maintenance, habitat, and gene pool protection | 42 | 22% |
| Water quality regulation | 20 | 10% |
| Water provision | 12 | 6% |
| Timber production | 10 | 5% |
| Cultural ES | 8 | 4% |
| Bioenergy production | 3 | 2% |
| Soil quality regulation | 3 | 2% |
| Natural hazard regulation | 2 | 1% |
| Genetic resources | 2 | 1% |
| Non-wood forest product production | 2 | 1% |
| Air quality regulation | 1 | 1% |

* One mechanism may target more than more ES.

Figure 2 shows the geographical distribution of the 88 MBIs identified: they spread across 34 countries covering six continents worldwide. The mechanisms are mainly located in North America (24%), Africa (22%), and Asia (21%). The top two countries with the highest number of cases are Uganda (10) and the United States of America (U.S.) (10). Two thirds (66%) of the mechanisms have been implemented in the long term (duration > 5 years), while 24% of those have been implemented in the short term (duration < 5 years) (Figure 3, left). More than half (55%) of the mechanisms have been implemented at the local level, while 26% at the national level (Figure 3, right).

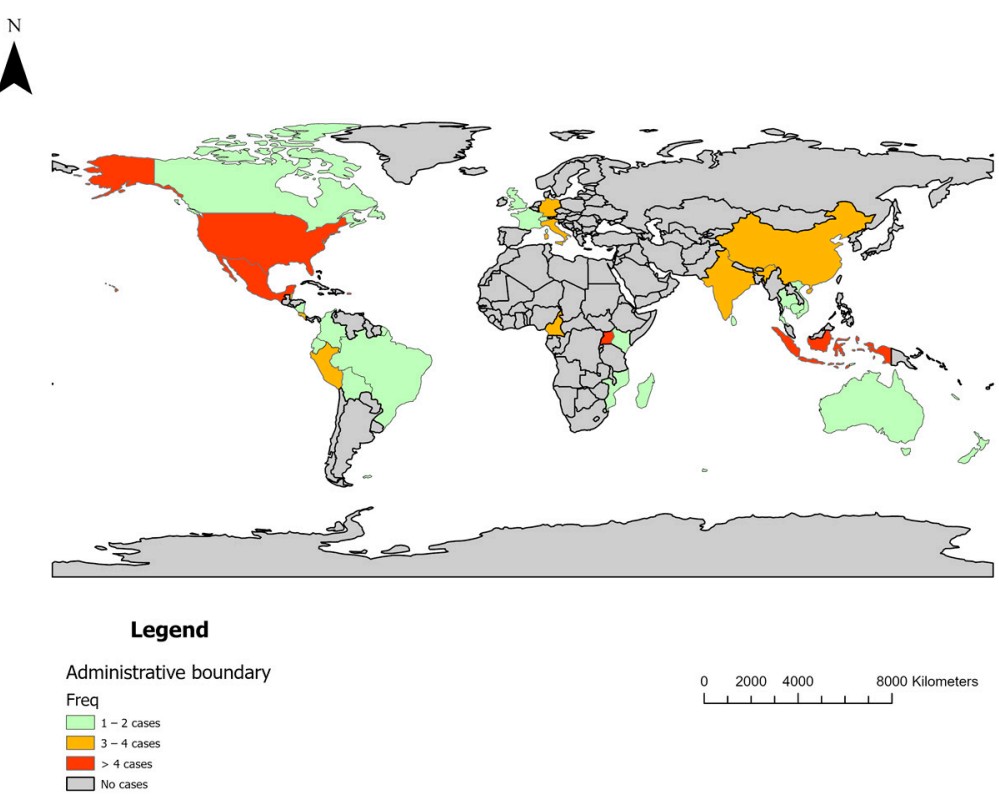

**Figure 2.** Geographical distribution of mechanisms reviewed (international mechanisms are excluded).

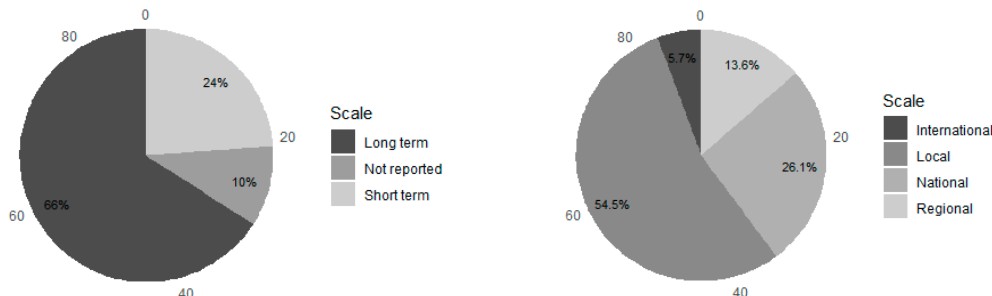

**Figure 3.** Left panel: temporal scale of mechanisms. Right panel: spatial scale of mechanisms.

Apart from environmental objectives, some mechanisms (30 out of 88) also have social objectives, such as improving local livelihoods (20), poverty alleviation (10), and promoting gender equity (1). There is one mechanism having two social objectives.

### 3.2. Typology of Market-Based Instruments for Forest Carbon

Nine types of MBIs are identified in the literature review, including PES and PES-like schemes, cap-and-trade, competitive tenders/auctions, definition of standards, certifications, eco-labelling, offset scheme, corporate social responsibility (CSR), public-private contract, subsidies and grants, and tax exemption and rebates (Tables 2 and 3). Among these, PES and PES-like schemes are the most frequently used MBIs. Detailed illustration for policy design features is shown in the Supplementary Material File S2.

**Table 2.** Description of policy design features for PES and PES-like schemes identified in the literature review (number in parentheses represents the number of mechanisms with corresponding features).

| MBI Types | Country | Spatial Scale | Funding Source | Financing Mechanism | Legal Source | Forest Ownership |
|---|---|---|---|---|---|---|
| PES and PES-like (REDD+) (29) | BZ (3) BO (1) BR (2) CM (2) KH (1) IN (1) ID (4) KE (1) LK (1) MG (1) MX (1) MZ (1) PE (3) TZ (3) UG (4) | National (1) Regional (4) Local (24) | Public (5) Private (9) Mixed (15) | Public payments (1) Public payments and voluntary transactions (1) Public payments and international aid/grant (1) Public payments and private funding (1) International aid/grant (3) International aid/grant and private funding (1) International aid/grant and voluntary transactions (8) Voluntary transactions and private funding (6) Voluntary transactions (5) Public payments, international aid/grant, and voluntary transactions (1) Public payments, private funding, and international aid/grant (1) | Supra-national and national law (1) Private or public law (contract) (28) | All forest land is owned publicly (5) Balanced ownership between public and private (2) Diverse forest ownership including public, private, and village ownership (4) Private ownership predominance (5) Public ownership predominance (13) |
| PES and PES-like (CDM) (8) | UG (4) IN (2) VN (1) CN (1) | Local (8) | Public (5) Private (2) Mixed (1) | Public payments (5) Voluntary transactions (1) Private funding (1) International aid/grant and voluntary transactions (1) | International law (8) | All forest land is owned publicly (1) Public ownership predominance (3) Private ownership predominance (4) |
| PES and PES-like (Joint implementation) (1) | CR (1) | Local (1) | Public (1) | International aid/grant (1) | International law (1) | Balanced ownership between public and private (1) |

Table 2. *Cont.*

| MBI Types | Country | Spatial Scale | Funding Source | Financing Mechanism | Legal Source | Forest Ownership |
|---|---|---|---|---|---|---|
| Government-led PES (17) | AU (1) CO (1) CN (2) CR (3) EC (1) MX (3) NZ (1) NI (1) PE (1) US (2) UG (1) | Local (3) Regional (3) National (11) | Public (16) Mixed (1) | Public payments (10) Public payments and international aid/grant (4) International aid/grant (2) Public payments and private funding (1) | Supra-national and national law (11) Private or public law (contract) (6) | Balanced ownership between public and private (3) Diverse forest ownership including public, private, and village ownership (4) Public ownership predominance (6) Private ownership predominance (4) |
| User-financed PES (13) | EC (1) FR (2) DE (1) ID (1) IT (1) MX (1) CH (1) TH (2) TL (1) UK (1) UG (1) | Local (9) Regional (2) National (2) | Private (6) Mixed (7) | Public payments and voluntary transactions (2) Voluntary transactions (7) International aid/grant (2) Public payments and private funding (2) Voluntary transactions and international aid/grant (1) | Supra-national and national law (3) Private or public law (contract) (10) | Balanced ownership between public and private (1) Diverse forest ownership including public, private, and village ownership (2) Public ownership predominance (3) Private ownership predominance (7) |

Table 3. Description of policy design features for market-based instruments identified in the literature review (number in parentheses represents the number of mechanisms with corresponding features).

| MBI Types | Country | Spatial Scale | Funding Source | Financing Mechanism | Legal Source | Forest Ownership |
|---|---|---|---|---|---|---|
| Cap-and-trade (5) | US (2) EU (1) CA (1) NZ (1) | National (4) Regional (1) | Public (3) Mixed (2) | Compliance transactions (5) | Supra-national and national law (5) | Private ownership predominance (3) Public ownership predominance (2) |
| Competitive tenders/auctions (7) | US (4) NZ (1) EU (1) AU (1) | National (6) Local (1) | Public (6) Private (1) | Compliance transactions (3) Public payments (3) Voluntary transactions (1) | Supra-national and national law (6) Private or public law (contract) (1) | Private ownership predominance (5) Public ownership predominance (2) |
| Definition of standards, certifications, eco-labelling (6) | US (4) CM (1) UK (1) | National (3) International (3) | Private (5) Mixed (1) | Voluntary transactions (5) Public payments and voluntary transactions (1) | Private law (private carbon standards) (5) Supra-national and national law (1) | Private ownership predominance (5) All forest land is owned publicly (1) |
| Offset scheme (53) | AU (2) VN (1) IT (2) DE (3) CH (1) CN (1) UK (1) BR (2) BZ (3) BO (1) KH (1) CM (2) CA (1) CR (1) FR (2) IN (3) ID (4) KE (1) MX (2) MZ (1) MG (1) PE (3) LK (1) TZ (3) TL (1) US (2) UG (8) EU (1) FR (2) TL (1) NZ (1) | Local (40) Regional (7) National (6) | Public (16) Private (18) Mixed (19) | Voluntary transactions (14) Public payments and voluntary transactions (2) Public payments and private funding (1) Public payments (7) Voluntary transactions and private funding (6) Compliance transactions (5) International aid/grant (5) International aid/grant and voluntary transactions (9) Public payment, international aid/grant, and voluntary transactions (1) Public payments and international aid/grant (1) Public payments and international aid/grant (1) Public payments, private funding, and international aid/grant (1) | Private or public law (contract) (37) Supra-national and national law (7) International law (9) | Private ownership predominance (18) Public ownership predominance (19) Balanced ownership between public and private (5) All forest land is owned publicly (6) Diverse forest ownership including public, private and village ownership (5) |
| Corporate social responsibility (7) | FR (2) IT (2) DE (1) TH (2) | Local (6) Regional (1) | Private (4) Mixed (3) | Voluntary transactions (6) Public payments and private funding (1) | Private or public law (contract) (7) | Private ownership predominance (4) Public ownership predominance (2) Diverse forest ownership including public, private, and village ownership (1) |

**Table 3.** *Cont.*

| MBI Types | Country | Spatial Scale | Funding Source | Financing Mechanism | Legal Source | Forest Ownership |
|---|---|---|---|---|---|---|
| Public-private contract (5) | IT (1) ID (1) UG (2) CR (1) | Local (4) Regional (1) | Mixed (2) Private (2) Public (1) | Public payments and private funding (1) Voluntary transactions and international aid/grant (3) Voluntary transactions (1) | International law (1) Private or public law (contract) (4) | Balanced ownership between public and private (1) Private ownership predominance (3) Public ownership predominance (1) |
| Subsidies and grants (2) | IN (1) AU (1) | National (2) | Public (2) | Public payments (2) | Supra-national and national law (2) | Public ownership predominance (2) |
| Tax exemption and rebates (1) | US (1) | National | Public (1) | Public payments (1) | Supra-national and national law (1) | Private ownership predominance (1) |

### 3.2.1. PES and PES-like Schemes

PES is a market-based approach to finance conservation activities, enhance ES provision and/or ecosystem quality, and support managers/owners with income opportunities. PES and PES-like schemes follow two principles: one is those who benefit from ES should pay for them, and those who provide ES should be compensated [37]. Sixty-nine out of 88 (78%) mechanisms apply PES and PES-like schemes. Among these, studies registered under institutional mechanisms were first identified and categorized, then the rest were classified based on their funding sources (i.e., public, private, or mixed).

Projects are mostly devoted to the REDD+ mechanism (29 out of 88 mechanisms), hence, at implementing activities to diminish human pressure on forests [38]. Project-based initiatives represent 83% of the total REDD+ projects identified in the review. International aid/grant is the most important funding channel, and nine REDD+ projects are funded by private funding sources. CDM is a market-based mechanism, in which countries with an emission reduction or emission-limitation commitment (Annex B Parties) need to implement an emission-reduction project in developing (non-Annex B) countries. Eight mechanisms registered under the CDM were identified. Joint Implementation (JI) is also a mechanism under the KP. In the JI, countries with emission reduction or limitation commitment under the KP (Annex B Parties) need to develop an emission-reduction or emission removal project in another Annex B Party. There is only one mechanism registered under the JI in this review.

PES and PES-like schemes can be distinguished between government-led PES and user-financed PES. Government-led PES is the PES scheme whose buyers are public bodies and funding sources for mechanisms are fully public [16]. User-financed PES is the PES scheme where funding directly comes from the beneficiary of the services provided but not from public bodies [16]. In the user-financed PES, funding sources for mechanisms are at least partially private or fully private. Seventeen mechanisms are devoted to the government-led PES and 11 of them are national-level programs. The funding is normally from public payments through state government revenue or international aid/grant (i.e., Global Environmental Facility (GEF), German and Norwegian governments). There are 13 user-financed PES mechanisms in the review and nine of them are at the local level. Six mechanisms are fully supported by private funding sources, while seven mechanisms are supported by both public and private funding sources. Voluntary transactions in the voluntary carbon market provide important income for project implementation. Buyers/demanders can be private companies, public institutions, public-private partnership, or non-governmental organizations (NGOs). For example, the buyer in the Mae Rim Watershed PES project, Thailand, is a private drinking water company, and the buyer in the Mae Taeng Watershed PES project is a parastatal water utilities company [39]. Federations, cantons, or municipalities act as buyers in the Agroforestry Climate project in Switzerland [40]. A foreign NGO is responsible for raising donations for the Carbon Forestry Scheme in Timor Leste [41].

### 3.2.2. Cap-and-Trade

Cap-and-trade is applied in 6% of the mechanisms. Cap-and-trade (also called emission trading scheme (ETS)) is a market-based approach in which entities that have emission units to spare can sell this excess capacity to entities that are over their emission targets. A cap is set on the total amount of GHG that can be emitted by the installations covered by the system and the cap is reduced along time so that total emissions go down [42]. The cap-and-trade scheme is usually implemented by governments and is backed up by regulatory frameworks. Notably, most existing cap-and-trade schemes take place in the developed countries, including U.S. (California Cap-and-trade), Canada (Forest Carbon Offsets Protocol), and New Zealand (New Zealand ETS).

### 3.2.3. Competitive Tenders/Auctions

Around 8% of mechanisms apply this MBI. Competitive tenders/auctions are the schemes that use a tendering process to allocate contracts to farmers or emission allowances to entities [16]. The use of auctioning is usually adopted together with cap-and-trade. In the emission trading, auctioning is used for allocating emission allowances and applying the 'polluter pays' principle [42]. For example, the Conservation Reserve Program (CRP) applied auctions to allocate land contracts to farmers to achieve environmental goals, such as improving water quality and preventing soil erosion [43]. Most mechanisms (6 out of 7) that used this instrument are national programs, and auction rules are normally set up by governments.

### 3.2.4. Definition of Standards, Certifications, and Eco-Labelling

Those mechanisms that used standards, certifications, and eco-labelling account for 7%. This instrument is aimed at providing information about product differentiation so that customers are willing to pay a higher value for products meeting certain standards [44]. The use of standards, certifications, and eco-labelling can be viewed from the perspective of supply chain. Standards can be understood as guidelines or criteria that it is mandatory to follow, which can be further classified into two categories: one focuses on products, and the other focuses on process. Certification is a procedure by which third-party organizations give a written assurance to either process or products [45]. Eco-labelling is a symbol showing the compliance with standards has been met, which provides an easy way for end-consumers to understand [45]. For example, the Kachung plantation, Uganda, is certified by the Climate, Community & Biodiversity Alliance (CCBA) for sustainable forest management [46]. In this case, CCBA is a process-type standard. Rainforest Alliance pays a price premium to cocoa farmers who grow shade trees in their cocoa orchards, so the cocoa in the orchards will be labelled as 'eco-friendly' [47]. Rainforest Alliance is a product-type standard. Four private carbon standards were established in the U.S., and they focus on certifying carbon projects and producing carbon credits. Certifications and eco-labelling are aimed at attracting financial support from private bodies for conservation activities. As shown in Table 3, funding sources for mechanisms under this category are at least partially private or fully private.

### 3.2.5. Offset Scheme

Offset scheme is applied in more than half (60%) of the mechanisms, which is also the second most frequently applied MBI for forest carbon in the review. The offset scheme is an instrument in which a loss in ES in one area is compensated by a similar gain in another [44]. In the scheme, ES users who create undesirable environmental impact are required to offset this impact by paying for conservation activities [44]. It is worth noting that PES is almost similar to the offset scheme except the latter is either driven by regulatory caps in the ETS or voluntary caps in the voluntary carbon market (VCM). As shown in Table 3, the offset scheme is commonly used in 31 countries and regions. Among these, 40 out of 53 mechanisms were implemented at the local level. Mechanisms in this category can be divided into compliance carbon offset projects (like ETS, CDM projects, Jurisdictional

REDD+ programs) and voluntary carbon offset projects (i.e., voluntary REDD+ projects). For example, a cap is set for GHG emissions in the EU ETS, and carbon offsets are allocated and traded among individuals or enterprises [48]. It is worth mentioning that forestry offsets have not been included in the EU ETS, but EU is taking steps to bring forestry into carbon market. Offset schemes can also be used with PES schemes, i.e., carbon offset contracts were given to forest landowners in a region in France to support carbon-oriented forest operations [49].

### 3.2.6. Corporate Social Responsibility

Mechanisms driven by corporate social responsibility (CSR) represent 8% of total mechanisms. CSR is a concept whereby companies integrate social and environmental concerns in the business operations and in their interaction with their stakeholders on a voluntary basis [50]. Besides CSR, environmental, social, and governance (ESG) is another concept that companies use to consider social and environmental outcomes with focus on material risks and opportunities [51]. Regarding environmental factors, ESG considers direct and indirect GHG emissions, corporates' stewardship of natural resources, etc. [51]. CSR and ESG are both driving forces behind VCM, and more and more companies are pledging to make contribution to climate change by reducing GHG emissions, such as Europ Assistance (the biggest private insurance company in Italy), AQUA (a large drinks manufacture), FLUVI (a water utilities company), etc. [39,40].

### 3.2.7. Public-Private Contract

A public-private contract is used in 6% of the mechanisms addressed by the review. It is a contract-based instrument between public (i.e., governments) and private sectors (i.e., private companies and forest landowners). The public-private contract can be set based on different activities. For example, in Uganda, a company obtained land leases from the state and got involved in the plantation forestry [46]. A public-private contract can also take form of a private company operating through public concession and acting as a carbon buyer in the upper Rio Reventazon watershed project, Costa Rica [52]. Concession is an attractive way to implement projects of public interest when public authorities need to mobilize private capital and the know-how to complement scarce resources [53]. Municipalities sometimes lack financial resources for creating green spaces, and public-private contract is the bridge joining municipalities with private companies. For instance, Europ Assistance (an Italian insurance company) funded tree planting activities in an urban park of Italy [40].

### 3.2.8. Subsidies and Grants

Two national-level mechanisms using subsidies and grants were identified in the review, which represent 2% of total mechanisms. 'Subsidies and grants are given to support forest holdings economically to encourage forest conservation or sustainable forest management aimed at enhancing or protecting biodiversity, soil, and water, securing recreational uses of the forest, climate regulation, and protection against natural hazards' [44]. Subsidies and grants can be passed from governments to forest owners/land managers or from higher levels of a country's government to lower levels. The Australia's Carbon Farming Initiative (CFI) incentivizes indigenous people to take sustainable farming practices (e.g., savanna burning) [54]. India's ecological fiscal transfer (EFT) is the grant from national governments to states or local governments [55]. National governments offer recurring payments from government revenue to lower-level governments based on their performance in achieving forest-related outcomes.

### 3.2.9. Tax Exemptions and Rebates

Only one mechanism in the review applies the tax exemptions and rebates instrument. Preferential forest property tax programs (PFPTPs) are applied in all the states of the U.S. PFPTPs are aimed at fostering ES by deferring, reducing, or eliminating property taxes

on enrolled private forest lands [56]. In this way, private forest owners are incentivized to plant and sustainably manage forests.

*3.3. Actors*

The Welfare Mix is introduced in this study to identify and categorize different actors involved in the mechanisms [57,58]. As shown in Figure 4, actor categories consist of state, community, market, and civil society. Four actor categories are distinguished on three criteria, namely (1) informal—formal, (2) for profit—non-profit, and (3) public—private [57]. Subcategories of state are intergovernmental organizations (e.g., Global Environmental Facility (GEF), United Nations Development Programme (UNDP)), government departments and agencies (three levels of governments), and donor agencies (e.g., international development agency of Norway and Denmark). Community consists of citizen and community groups (i.e., private forest owners, land managers, community associations) and indigenous peoples and local community (i.e., indigenous communities). Market category consists of a private sector, including companies or organizations who have carbon offsetting needs, carbon certification bodies, carbon brokers, forestry companies, etc. Civil society can be divided into NGOs (e.g., WWF, Clinton Climate Initiative) and science and educational organizations (e.g., University of California, Mexican National Institute of Ecology). It also includes many intermediary organizations/institutions that cross the boundaries between profit and non-profit; private; and public, formal, and informal [57].

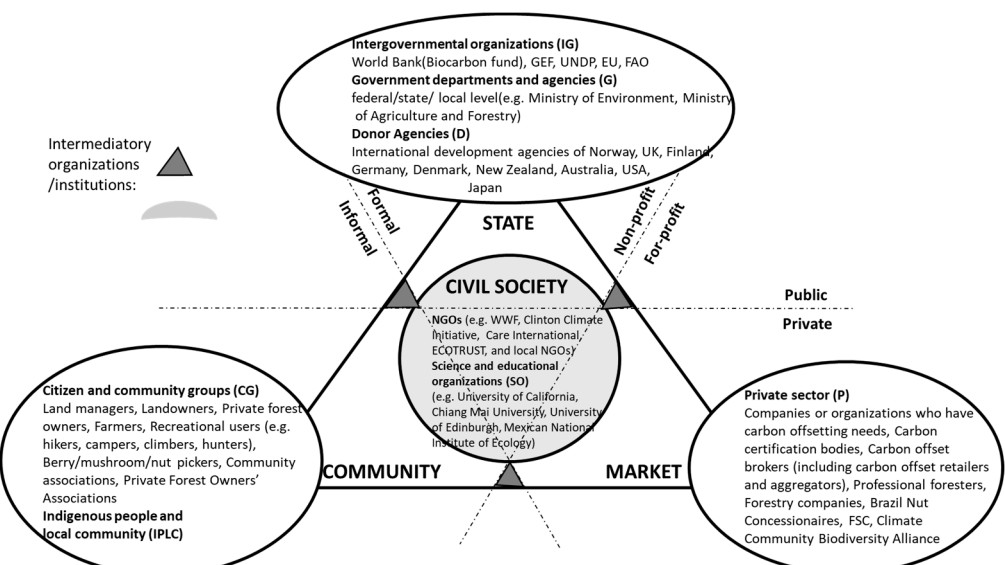

**Figure 4.** Actor categories identified in the review from a multi-actor perspective (adapted from Avelino & Wittmayer [57] and IPBES [59]).

The most common service buyers/demanders are actors under state and market categories (See Table S2). Governments (G), intergovernmental organizations (IG), and donor agencies (D) each act as service buyers/demanders in 35%, 18%, and 17% of total mechanisms in the literature. The common IGs identified as buyers/demanders are the World Bank, EU, GEF, UNDP, etc. International development agencies usually provide start-up funding to mechanisms, such as United States Agency for International Development (USAID), Norway's International Climate and Forest Initiative (NORAD), Australia's development program (AUSAID). More than half (58%) of the mechanisms involve a private sector (P) as buyers/demanders. Among these, companies or organizations who have carbon offsetting needs are the predominant buyers/demanders. These companies are either acting under regulatory obligations or are driven by CSR, including energy corporations, water utility companies, public transport companies, spring water bottling plants, conference, festivals, etc. The community sometimes behaves as a service buyer/demander,

which represents 10% of mechanisms. Under this category, there are service end-users (i.e., fuel taxpayers), recreational users (i.e., tourists, hikers), landowners, farmers etc. NGOs act as buyers/demanders in 8% of the mechanisms reviewed, including international NGOs (i.e., The Nature Conservancy, Clinton Foundation, CARE International), national NGOs (e.g., Fundacion Amigos de la Naturaleza from Bolivia), and local NGOs.

Community is the most common service seller/provider. Land tenure is not clear in some studies, and farmers and landowners are used interchangeably. Therefore, landowners, farmers, land managers, and private forest owners are attributed to the same category—citizens and community groups. CG and IPLC are involved in 60% and 36% of total mechanisms, respectively. Local communities usually participate through community-based organizations, such as community forestry groups and community associations. Governments act as sellers/providers in 11% of total mechanisms. The private sector participates as a seller/provider through obtaining licenses from the state in the form of land leases, which account for 9% of mechanisms.

A large majority (80%) of mechanisms in the literature review involves intermediaries/facilitators. The most common ones identified in the mechanisms are government departments and agencies, which work as intermediaries/facilitators in 41% of total mechanisms, such as National Forestry Commission (CONAFOR) in Mexico, National Forest Authority (NFA) in Uganda, Clean Energy Regulator in Australia, etc. NGOs also play an important role in facilitating and bridging. Approximately 31% of total mechanisms involve NGOs as intermediaries/facilitators. Mechanisms that involve private sector intermediaries represent 28% of the total. Private sector intermediaries include carbon certification organizations, carbon brokers, private trust funds (i.e., Althelia Funds in Peru), tree planting companies, private logging companies, etc. [60]. Science and educational organizations (13%) are mainly in charge of providing technical support.

*3.4. Cluster Analysis*

Mechanisms extracted from the literature show differences across funding sources, scales, geographic locations, actors involved, the use of MBIs, etc. MCA and cluster analysis were performed to identify underlying patterns. The analysis was conducted on an indicator matrix, where the rows represent mechanisms (88), and the columns are variables representing attributes of mechanisms (17). The variables included in the analysis are continents, spatial scales, funding sources, forest ownerships, and the presence of MBIs (see Table S3). The indices computed in the MCA were further used for cluster analysis, including eigenvalues, variances explained of principal components (PCs), and factor loadings (see Table S4). Factor loadings are the correlation coefficients between the variables and the PCs. There is a substantial drop in all indicators between the 5th and the 6th PCs. Therefore, five components were kept for analysis to make interpretation easier, which accounts for 45.33% cumulative variance. Each dimension represents a combination of variables, and it stands for a proportion of total variability. Figure 5 shows the first two dimensions of MCA as an example, which account for a 23.13% of the total variability of 88 mechanisms per 17 variable matrices. Objects with the closest coordinates are similar to each other and objects in the same cluster are more similar to each other than those in other clusters. Inertia measures how well a dataset is clustered by K-means, and it can be seen from Figure 5 that three clusters is a reasonable trade-off between relatively low inertia and few clusters [61].

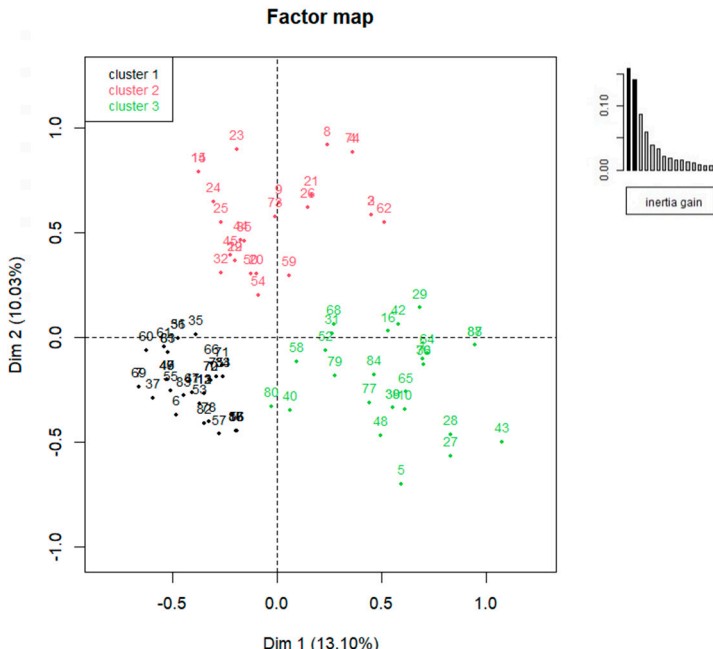

**Figure 5.** Cluster analysis of mechanisms reviewed in the literature (points represent mechanisms, and inertia gain shows how much variance will be obtained by adding more clusters).

- Cluster 1 is labelled as 'legally binding mechanism', which is defined by the characteristics as follows: mechanisms registered under REDD+, CDM, or JI; applying the offset scheme; local scale; and mechanism is implemented in Africa.
- Cluster 2 is labelled as 'private standard and user-financed PES', which is defined by the characteristics as follows: funding source of mechanism is private; applies the definition of standards, certifications, eco-labelling, or user-financed PES in mechanism; driven by corporate social responsibility; international scale; mechanism is implemented in Europe; and private ownership predominance in forest land.
- Cluster 3 is labelled as 'public cap-and-trade and government-led PES', which is defined by the characteristics as follows: funding source of mechanism is public; applies government-led PES schemes, cap-and-trade, or auctions; and national scale.

Entities in cluster 1 are closer to each other than those in cluster 2 or 3, which indicates that forest carbon mechanisms in the 'legally binding mechanism' category are more similar to each other in terms of mechanism attributes, i.e., continents, spatial scales, funding sources, forest ownership, etc.

## 4. Discussions

This study, based on a corpus of 87 relevant references from peer-reviewed journals, provides an overview of diverse MBIs used in forest carbon projects. Benefiting from the bloom of MBI literature and continuous addition of new case studies, a global and up-to-date database of mechanisms using MBI has been constructed by a systematic literature review.

### 4.1. Conceptual Model

A conceptual model was developed to provide clarity for interpreting results. This model was inspired from empirical evidence of MBIs reported in the literature and social-ecological system (SES) framework [62–64]. Figure 6 shows the key components of governance of MBIs for forest carbon and their relationships. Through this conceptual model, findings in the isolated case studies can be organized to provide cumulated evidence on this topic. The SES framework was originally designed for being applied to common-pool resource management [65].

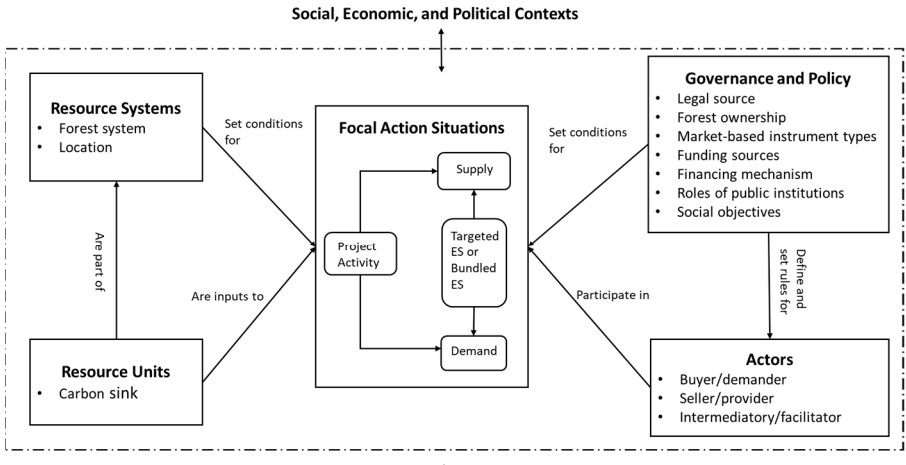

**Figure 6.** Conceptual model for governance of market-based instruments for forest carbon (adapted from McGinnis & Ostrom [65]).

As in the SES framework, our conceptual model also contains five segments, including resource systems, resource units, focal action situations, governance and policy, and actors. As shown in the systematic literature review, these segments interact at different spatial (i.e., local, subnational, national, and international) and temporal scales (i.e., short-term and long-term). More than half of mechanisms using MBI are at local level, and one of them (Trees for Global Benefits, Uganda) was mentioned seven times in the review. We can expect that some of the local mechanisms can provide enough information to scale-up and replicate successful experiences. Different social, economic, and political contexts affect the interaction of these segments, such as economic development, demographic trends, and political stability. The resource system is limited to forest system in this study. The resource unit is described by a single element, carbon sink.

Focal action situations describe how project activity affects the supply and demand chain of single or bundled ES. Among these, afforestation/reforestation, improved forest management, and reducing emissions from deforestation and forest degradation are the most common project activities, which reconfirms the previous funding by Chenost et al. [66].

Ecosystem services refer to targeted ES in the mechanism. In this article, particular attention has been given to carbon sequestration and storage, but some mechanisms targeting hydrological ES (i.e., water quality regulation and water provision) were included in the review as they also provide co-benefits related to carbon, which is consistent with previous findings that there is a trend towards concentrating on the most valuable ES and using these as 'umbrella services' to achieve a range of conservation goals [67].

Governance and policy are the focus of this study, which encompass the governance system and policy design. Legal sources and forest ownership describe institutional and legal contexts of forest carbon mechanisms. Instrument types refer to MBI typologies identified in the literature (see Section 4.2). Financing mechanism and funding sources specify institutional design features of forest carbon mechanisms, and many of them applied a diversity of funding and revenue sources to ensure long-term program sustainability. Actors specify who participates in exchanging services and goods, including buyers/demanders, sellers/providers, and intermediaries/facilitators (see Section 4.4). Many forest carbon mechanisms in this review have social objectives, through which they connect social needs to ES, such as poverty alleviation, improving local livelihoods and gender equity. This is aligned with the statement made by Sarira et al. [68] that 'forest carbon projects can deliver multiple benefits to society.' The role of public institutions is discussed in Section 4.5.

### 4.2. The Diverse Typology of Market-Based Instruments

In line with study by Pirard and Lapeyre, the classification of MBIs for the forest carbon sink is highly fragmented, diverse, and complex [3]. By introducing a conceptual model and

uniform terminologies, we established the common ground for comparing policy instruments implemented in different contexts. In this analysis, nine types of MBIs were analyzed, including PES and PES-like schemes, cap-and-trade, competitive tenders/auctions, definition of standards, certifications, eco-labelling, offset scheme, corporate social responsibility, public-private contract, subsidies and grants, and tax exemption and rebates.

Clear and secure forest tenure rights play an important role in the successful implementation of MBIs, especially REDD+ [69]. Various types of forest land tenure have been identified, and tenure insecurity has also been found in some of the forest carbon mechanisms reviewed in this study. For instance, there have been overlapping claims on village land by concession holders in the Ketapang Community Carbon Pools project [70]. Another example is the Rimba Raya Biodiversity Reserve Project in which palm oil companies and communities have some primary tenure issue over land [70]. Tenure insecurity at the project level is also linked with national arrangements. Both examples are from Indonesia where governments fail to recognize community customary rights in forests. Besides, the tenure issues are rooted deeply in land use history. In this case, external help from public authorities (i.e., local and/or provincial or state government) is usually necessary to address tenure challenges through conducting community mapping and spatial planning, clarifying legal ownership of forest carbon, recognizing community forest rights, etc. [70].

Through MCA and cluster analysis, we identified and labeled three distinctive groups as follows: 'legally binding mechanism', 'private standards and user-financed PES', and 'public cap-and-trade and government-led PES'. Each group has a set of attributes (e.g., the use of policy instruments, forest tenure, geographical location). The cluster analysis results are consistent with previous attempts that there are distinctive differences between public and private PES schemes [12,43]. Under the context of forest carbon projects, another cluster is identified as 'legally binding mechanism', which are mechanisms registered under international framework convention on climate change, such as the KP. Due to increasing cooperation against climate change, mechanisms registered under the international framework convention were attributed into one group. The use of policy instruments could be determined by forest ownership. Our inventory has shown that the application of user-financed PES schemes is associated with private ownership predominance in forest land, while government-led PES schemes are more likely to take place in countries with public ownership predominance in forest land.

### 4.3. Policy Mix and Their Interactions

We categorized mechanisms based on their uses of policy instruments. Our results have documented that there may be multiple policy instruments coexisting in the same country, region, or even mechanism. Pirard and Lapeyre also stated in their study that public policies are more likely to combine rather than choose between different policy instruments [3]. In order to describe their interactions, the concept of policy mix was applied. Ring and Schröter-Schlaack [13] defined a policy mix as a combination of policy instruments that has evolved to influence the quantity and quality of ecosystem service provision in both public and private sectors.

The interaction between policy instruments can be complementary, and one example combines government-led PES scheme with pre-existing command-and-control polices [71]. The Natural Forest Conservation Program conserves natural forests in China by imposing logging bans and offering forest enterprises incentives to conduct afforestation activities [72]. A similar approach can be observed in the Bolsa Floresta program, Brazil. The PES initiative was introduced on top of already existing regulations for local sustainable development reserves [73]. Börner et al. modelled the impact of mixing 'carrots' (in this case, PES) and 'sticks' (e.g., a regulatory enforcement strategy) to conserve forests in the Brazilian Amazon, and they found out introducing PES increases policy implementation costs, reduces income losses for those hit the hardest by law enforcement, meanwhile provides additional income to some land users [74]. Another reason for the use of policy mixes is to improve administrative capacity [75]. For example, Indian EFT is imposed

on the pre-existing conservation policies, which helps to close gaps between different administrative levels by internalizing conservation costs and benefits of protected areas that spill over administrative boundaries of lower-level public administrations [14,55]. The policy interactions indicate that MBIs are embedded in the broader social and political relationships that already established among actors, and these relationships need to be acknowledged when introducing new MBIs [21].

Information instruments are sometimes applied simultaneously with MBIs as well. For instance, corporations driven by CSR, acted as buyers/demanders in the Sylv'ACCTES program, France, and they voluntarily paid landowners to change their land management practices in ways that deliver ES [76]. Meanwhile, leaflets were distributed to consumers creating a widespread perception towards environmental problems [49]. The use of information instruments prepared target groups with relevant knowledge that may improve the outcome of MBIs [77]. We also found that PES and PES-like schemes tend to be combined with standards and certifications. For example, a REDD+ project (i.e., the Bukaleba plantation), funded by international donors, is also certified according to the Forest Stewardship Council (FSC) standards, and validated and verified under Verified Carbon Standard (VCS) [46]. In this way, it develops diverse financing sources for start-up and project implementation, making the mechanism run sustainably.

Our results have shown that interactions can also be interdependent, which means some policy instruments rely on interactions with other instrument types to achieve their policy objectives [78]. Cap-and-trade, competitive tenders/auctions, and the offset scheme are usually applied simultaneously, and they depend on each other. For example, cap-and-trade provides a regulatory framework for exchanging carbon offsets. Meanwhile, competitive tenders/auctions are used to distribute carbon offsets to firms having carbon emission needs for cost-effectiveness.

### 4.4. Actor Roles from a Multi-Actor Perspective

Governance of MBIs for forest carbon is a multi-actor process that incorporates socio-economic, institutional, and political elements [79]. By using the Welfare Mix and uniform terminologies, it synthesizes actors involved in mechanisms and enable us to compare actor roles in different contexts. Based on the Welfare Mix, actors were divided into four categories: state, community, market, and civil society.

Our results have shown that governments, international development agencies, and the private sector play important roles as service buyers/demanders. Community is the most common service seller/provider. Aligned with findings by Martin-Ortega et al. [32], a large majority of mechanisms reviewed in this analysis involves intermediaries/facilitators, mostly NGOs and governments. This can be explained by the fact that intermediaries can decrease transaction costs [1]. However, it is worth noting that they can also increase transaction costs in some cases. For example, private sector intermediaries act as carbon brokers in the projects, and they reduce benefits getting back to forest owners/managers and to forests. Science and educational organizations play a facilitator role by providing technical support to mechanisms.

It is worth mentioning that NGOs play an important role in forest carbon projects, which is consistent with previous findings by Shin et al. [80]. For example, in the Trees for Global Benefits project, the NGO intermediatory is responsible for facilitating sales of carbon offsets and signing agreements on behalf of the buyers [81]. The nature conservancy (U.S.-based environmental NGO) worked as carbon offset validation and verification technical advisors in the Rio Bravo climate action project, Belize [82]. A local NGO designed and implemented a PES program in Uganda [83]. Many NGOs can also empower local communities, which is essential for community-based projects [80].

State-led or market-logic mechanisms have prevailed for decades. Notably, a new surge in community-based mechanisms can be observed in the literature [57]. The community-based approach is aimed at enhancing the engagement and inclusion of indigenous people and forest communities in policy implementation and project activities [84]. Eight mech-

anisms in this review are community-driven and most of the mechanisms are registered under the REDD+ or CDM. It can be expected that the state is increasingly calling upon 'the community' to govern public goods, and governments are restructuring their responsibilities and jobs about their citizens [57]. However, there are also some signals suggesting the opposite direction, for example, the European Commission defined standards in the carbon farming initiative [85].

*4.5. The Pronounced Role of Public Institutions*

Although forest carbon is normally organized through a market-oriented way, our study has reconfirmed the previous finding that the role of public institutions is still pronounced and unneglectable [1,86]. This can be explained by the increasing recognition that MBIs do not operate alone but in the 'hybrid governance' model (i.e., public administrations and private actors are no longer at the opposite sides, and they can be aligned around a shared interest) [16,71,87]. The role of public institutions in three clusters is identified and summarized in Table 4.

**Table 4.** Role of public institutions in each cluster identified in the analysis.

| The Role of Public Institutions | Cluster of MBI Mechanisms | | |
|---|---|---|---|
| | Cluster 1 | Cluster 2 | Cluster 3 |
| Defining the liabilities (cap) and overseeing those are respected | | ✓ | |
| Setting up policy, legal and regulatory frameworks | ✓ | ✓ | |
| Providing funding for project implementation | ✓ | ✓ | ✓ |
| Providing technical support to facilitate project activities (i.e., seedlings, advice, holding workshops) | ✓ | ✓ | ✓ |
| Acting as 'carbon aggregator' | ✓ | ✓ | |
| Acting as service buyers/demanders | | ✓ | ✓ |
| Acting as service sellers/providers | ✓ | ✓ | |
| Acting as an intermediatory between different organizations | ✓ | ✓ | |
| Acting as administrators of mechanism | ✓ | ✓ | |
| Initiating public–private partnership | | | ✓ |
| Impeding project implementation | ✓ | | |

Public institutions play an important role in Cluster 1—'legally binding mechanism'. For example, Brazilian REDD+ strategy took a jurisdictional approach (i.e., carbon monitoring will occur over an entire political administrative region) and exhibited strong subnational government leadership, where public bodies acted as project implementors and intermediaries/facilitators [88,89]. The REDD+ strategy began with an international organization and ran 'down' through the national, regional, and local governments in Brazil [90]. In the Ngoyla-Mintom REDD+ project, the Cameroonian government facilitated REDD+ implementation by setting up a regulatory framework in which local people were allowed to create legal entities and manage forest locally [91]. Benjaminsen and Kaarhus [92] gave evidence from Tanzania about a national organization acting as a 'carbon aggregator' to help local communities reduce high opportunity costs. Wells et al. [81] and Shames et al. [93] showed that public administrations also provide technical support to facilitate project activities, e.g., providing seedlings, giving advice on pests and diseases, or holding training workshops. In some cases, public administrations might impede the

implementation of mechanisms, for example, the Indian government required NGOs to go through a complex approval process for the Khasi Hills Community REDD+ project [94].

The role of public institutions is predominant in Cluster 2—'public cap-and-trade and government-led PES'. In the cap-and-trade, public administrations mainly act as regulators in defining the liabilities (cap) and overseeing those are respected. They enable legislations for ETS and define institutional arrangements related to that. Further operational specifications in the cap-and-trade program are also defined by public administrations. Governments can act as carbon aggregators and they purchase carbon offsets from service providers and sell the carbon offsets on the market, such as the Canadian government in Forest Carbon Offsets Protocol (FCOP) [95]. Governments also take a role in providing funding for the implementation of subsidies and grants through their revenue, such as India's EFT [55]. Governments also defer, reduce, or eliminate property taxes on private forest lands. In the offset scheme, public organizations, particularly national park administrations, sometimes act as service sellers/providers since the forests are public [39]. The role of public institutions is dominant in government-led PES programs, and they usually act as buyers, facilitators, and administrators. For example, the Chinese government acted as buyer in the Grain for Green Program and provided incentives (both money and in-kind benefits) to rural households [96]. The state forest agency of Mexico is responsible of channeling the funding for the local matching funds PES [97]. Montoya-Zumaeta et al. reported that the Peruvian government granted collective or individual land rights to communities to facilitate the implementation of the National Forest Conservation Program [60]. Governments are responsible for the administrative functions of Payment for Environmental Services Program in Costa Rica [98].

The role of public institutions is not so noticeable in Cluster 3—'private standards and user-financed PES'. Even in the user-financed PES schemes, financing sources are not fully private, instead blended finance is used. Blended finance is entitled as a model for financing development projects that combine an initial investment (usually from governments) with additional private capital [99]. For instance, the UK's Peatland Code is developed with technical and monetary support from governments, meanwhile it is developed as a voluntary certification standard for peatland projects seeking additional private funding via the voluntary carbon market [100]. Regional governments in Germany provided start-up funding for the Moorfutures project [101]. Local and regional authorities in France also participated in a user-financed PES mechanism as buyers to reinforce their forest-based policies [49]. Public organizations also play a role in public-and-private contracts. In the Rimba Raya Biodiversity Reserve Project, a revenue-share program was established between the Management Authority of the National Park and local communities with the help of governments [102].

It is worth noticing that the role of public institutions in these instruments evolves with the policy development rather than remaining constant across time. For example, the potential inclusion of forestry into the EU ETS was dismissed in 2003; however, the New EU Forestry Strategy, in 2021, claimed that forest investments will be included in the Carbon Farming Initiative (an initiative that is aimed at generating carbon offset to be sold in the ETS) [103]. Under this context, it is clear that public authorities will play a more crucial role in the increasingly institutionalized forest carbon market. Another example is represented by the finalization of Article 6 of the Paris Agreement, which allows countries to reach their Nationally Determined Contribution (NDC) targets through participation in the global carbon market [104]. In other words, a country will be able to transfer carbon credits earned from mitigation projects to help another country (or countries) meet climate targets [104]. Carbon offsets can be generated by private actions through a crediting mechanism nested within transnational partnerships [105]. The new role of public authorities under this policy development includes developing and prioritizing areas for transnational partnership engagements, defining minimum criteria and procedural requirements for non-state actors, actively supporting strategic initiatives, facilitating market or non-market finance, and evaluating the effectiveness of partnerships [105]. The 'nested' approach with high-level

state structure involvement will provide the coercion and other resources that make local negotiation efficient [106]. Also, it emphasizes the importance of decision making at multiple levels and provides concrete arenas where local actors can have a say [107].

*4.6. Future Research*

This systematic literature review provides a framing for understanding the empirical varieties of policy configurations related to forest carbon. As stated in the methodology, we undertook data collection only from the Scopus database and did not include reports from international organizations (e.g., Center for International Forestry Research), which may affect our findings. Besides, we only focused on literature written in English, which may fail to be fully representative of research in the global south. Needless to say, the mechanisms reviewed in our study are not fully exhaustive of all the possible existing mechanisms using MBI for forest carbon. A future study should involve more case studies, for example, by considering different citation databases. Only a few variables were contained in the cluster analysis, and this can be enhanced by including more socio-economic or biophysical variables. The concepts of policy mix and policy instrument interaction were introduced in the study to explain the combination of policy instruments applied in one mechanism. Future research can be done by exploring the relationship between the emergence of policy mixes and institutional or governance contexts. Also, the effectiveness of the policy mix for forest carbon needs further evaluation.

**5. Conclusions**

Forest carbon plays an important role in mitigating climate change. The meta-analysis carried on in this study provides an overview of MBIs for forest carbon and compares different policy options. Based on published literature and adopting a systematic literature review approach, a conceptual model is developed to provide clarity about the governance of MBIs. The market for forest carbon has proven to be heterogenous, and nine types of MBIs are identified in the review. Through cluster analysis, forest carbon mechanisms are divided into three groups and labelled as 'legally binding mechanism', 'private standard and user-financed PES', and 'public cap-and-trade and government-led PES'. The theory of policy mix and interaction is applied, and we find that the interplay among policy instruments in the projects can be complementary or interdependent. With the analytical lens of hybrid governance, our results indicate that public institutions play diverse roles in forest carbon projects, and public funding is still the main financing source. Given the complex interplay of these various policy instruments, there are key implications for policy makers. First, MBI alone is never a silver bullet, rather it works the best with other policy instruments. We highlight the importance of understanding real-world policy challenges from the perspective of policy mix and suggest shifting away from assessing the efficacy of individual MBI. Second, public institutions are going to play an increasing role in the future climate policy arena (e.g., the Paris Agreement), so experts and policy dialogues can help to recognize the changing role of them under policy development. Third, we suggest public institutions should stand out in the market-driven mechanism and create more attractive conditions for private governance and finance in the forest sector. For example, governments can improve smallholders' market access to forest carbon through providing provisional support and they can also design a more effective carbon pricing mechanism, attracting more people to invest.

**Supplementary Materials:** The following supporting information can be downloaded at: https://www.mdpi.com/article/10.3390/f14010136/s1. Supplementary Material File S1, Figure S1: Temporal distribution of establishment of mechanisms; Table S1: Search string of two-round literature search; Table S2: Frequency and proportion of actors involved in the reviewed mechanisms (in some cases, there are multiple buyers or intermediaries participating in the mechanisms); Table S3: Description of variables used in the cluster analysis; Table S4: Principal Component (PC) extraction (eigenvalues and variance explained of the first seven PCs) and factor loadings (only loadings of the first five PCs are reported); Supplementary Material File S2: Inventory of forest carbon mechanisms using market-based instruments.

**Author Contributions:** Conceptualization, X.S., P.G., and F.P.; methodology, X.S. and F.P.; validation, X.S.; formal analysis, X.S.; writing—original draft preparation, X.S.; writing—review and editing, X.S., P.G., and F.P.; supervision, P.G. and F.P. All authors have read and agreed to the published version of the manuscript.

**Funding:** This research received no external funding.

**Institutional Review Board Statement:** Not applicable.

**Informed Consent Statement:** Not applicable.

**Data Availability Statement:** See Supplementary Materials.

**Acknowledgments:** The authors would like to thank Mauro Masiero for his valuable help.

**Conflicts of Interest:** The authors declare no conflict of interest.

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
