# Peer review of "Unravelling the Role of Institutions in Market-Based Instruments: A Systematic Review on Forest Carbon Mechanisms"

_forests, doi:10.3390/f14010136_

Round 1

Reviewer 1 Report

I enjoyed reading this well-written article. The objectives are clearly defined. 

Major concerns:

1. The forest policies described in the paper are not all market-based instruments. In environmental economics, the MBI include PES, taxes, subsidies, and cap-and-trade programs (see environmental economics textbooks). Some of policies listed in this review do not have a market component (e.g. education campaigns, logging bans). The paper should change MBIs to environmental policies (or forest policies) or at a minimum explain why the other mechanisms are considered MBI. In section 3.2.10, the paper acknowledges that CC and IE are not part of MBIs, but they are part of your review. Thus, you may consider changing the focus of the paper and title to be based on forest policies. 

2. What's the difference between a PES and offsets? Isn't there an overlap? There could be a corporation, NGO, or government paying for an ecosystem service as an offset. 

3. Actors (section 3.3). It would be better to find a more descriptive name for the third sector (e.g. non-profits) that is analogous to the other sectors. 

4. Figure 4 is hard to read. Is the circle of the third sector supposed to be contained within the triangle such that the circle is within formal and non-profit lines?

5. The paper will benefit from some discussion about land tenure structures in the meta-analysis. Are there formal/informal? Are there communal land tenure structures in the area? The latter analysis will enhance the metanalysis. 

6. What are measures of goodness of fit within the clusters? In Figure 5, cluster 1 have closer points relative to clusters 2 and 3. Comments about the cluster quality should be included

7. Limitations: within the section of future research, limitations of this article should be specified.

- For example, does focusing on articles written in English result in a sample selection that does not offer a wider representation of research in the global south. 

- This paper doesn't include reports from key organizations such as the CIFOR (https://www.cifor.org/). This should be acknowledge at some point. 

8. The following peer-reviewed articles from top environmental and natural resource economics journals are missing from this review:

Alix-Garcia, J., & Wolff, H. (2014). Payment for ecosystem services from forests. Annual Review of Resource Economics, 6(1), 361–380.

Angelsen, A., & Rudel, T. K. (2013). Designing and Implementing Effective REDD + Policies: A Forest Transition Approach. Review of Environmental Economics and Policy, 7(1), 91–113. https://doi.org/10.1093/reep/res022

Baragwanath, K., & Bayi, E. (2020). Collective property rights reduce deforestation in the Brazilian Amazon. Proceedings of the National Academy of Sciences, 117(34), 20495–20502. https://doi.org/10.1073/pnas.1917874117

Barbier, E. B., & Tesfaw, A. T. (2013). Tenure constraints and carbon forestry in Africa. American Journal of Agricultural Economics, 95(4), 964–975.

Blackman, A., Corral, L., Lima, E. S., & Asner, G. P. (2017). Titling indigenous communities protects forests in the Peruvian Amazon. Proceedings of the National Academy of Sciences, 114(16), 4123–4128. https://doi.org/10.1073/pnas.1603290114

Bluffstone, R. (2013). Economics of REDD+ and community forestry. Journal of Forest and Livelihood.

Busch, J., & Ferretti-Gallon, K. (2017). What Drives Deforestation and What Stops It? A Meta-Analysis. Review of Environmental Economics and Policy, 11(1), 3–23. https://doi.org/10.1093/reep/rew013

Christensen, D., Hartman, A. C., & Samii, C. (2021). Citizen monitoring promotes informed and inclusive forest governance in Liberia. Proceedings of the National Academy of Sciences, 118(29), e2015169118. https://doi.org/10.1073/pnas.2015169118

Eisenbarth, S., Graham, L., & Rigterink, A. S. (2021). Can community monitoring save the commons? Evidence on forest use and displacement. Proceedings of the National Academy of Sciences, 118(29). https://doi.org/10.1073/pnas.2015172118

Ferraro, P. J., & Agrawal, A. (2021). Synthesizing evidence in sustainability science through harmonized experiments: Community monitoring in common pool resources. Proceedings of the National Academy of Sciences, 118(29), e2106489118. https://doi.org/10.1073/pnas.2106489118

Ickowitz, A., Sills, E., & de Sassi, C. (2017). Estimating Smallholder Opportunity Costs of REDD+: A Pantropical Analysis from Households to Carbon and Back. World Development, 95, 15–26. https://doi.org/10.1016/j.worlddev.2017.02.022

Ituarte-Lima, C., McDermott, C. L., & Mulyani, M. (2014). Assessing equity in national legal frameworks for REDD+: The case of Indonesia. Environmental Science & Policy, 44, 291–300. https://doi.org/10.1016/j.envsci.2014.04.003

Kansanga, M. M., & Luginaah, I. (2019a). Agrarian livelihoods under siege: Carbon forestry, tenure constraints and the rise of capitalist forest enclosures in Ghana. World Development, 113, 131–142. https://doi.org/10.1016/j.worlddev.2018.09.002

Liscow, Z. D. (2013). Do property rights promote investment but cause deforestation? Quasi-experimental evidence from Nicaragua. Journal of Environmental Economics and Management, 65(2), 241–261. https://doi.org/10.1016/j.jeem.2012.07.001

Lubowski, R. N., & Rose, S. K. (2013). The Potential for REDD+: Key Economic Modeling Insights and Issues. Review of Environmental Economics and Policy, 7(1), 67–90. https://doi.org/10.1093/reep/res024

Lund, J. F., Sungusia, E., Mabele, M. B., & Scheba, A. (2017). Promising Change, Delivering Continuity: REDD+ as Conservation Fad. World Development, 89, 124–139. https://doi.org/10.1016/j.worlddev.2016.08.005

Oldekop, J. A., Sims, K. R. E., Karna, B. K., Whittingham, M. J., & Agrawal, A. (2019a). Reductions in deforestation and poverty from decentralized forest management in Nepal. Nature Sustainability, 2(5), 421–428. https://doi.org/10.1038/s41893-019-0277-3

Ollivier, H. (2012). Growth, deforestation and the efficiency of the REDD mechanism. Journal of Environmental Economics and Management, 64(3), 312–327. https://doi.org/10.1016/j.jeem.2012.07.007

Pfaff, A., Amacher, G. S., & Sills, E. O. (2013a). Realistic REDD: Improving the Forest Impacts of Domestic Policies in Different Settings. Review of Environmental Economics and Policy, 7(1), 114–135. https://doi.org/10.1093/reep/res023

Ravikumar, A., Larson, A., Duchelle, A., Myers, R., & Tovar, J. G. (2015). Multilevel governance challenges in transitioning towards a national approach for REDD+: Evidence from 23 subnational REDD+ initiatives. International Journal of the Commons, 9(2), 909–931. https://doi.org/10.18352/ijc.593

Simonet, G., Subervie, J., Ezzine-de-Blas, D., Cromberg, M., & Duchelle, A. E. (2019). Effectiveness of a REDD+ Project in Reducing Deforestation in the Brazilian Amazon. American Journal of Agricultural Economics, 101(1), 211–229. https://doi.org/10.1093/ajae/aay028

Slough, T., Rubenson, D., Levy, R., Alpizar Rodriguez, F., Bernedo del Carpio, M., Buntaine, M. T., … Zhang, B. (2021b). Adoption of community monitoring improves common pool resource management across contexts. Proceedings of the National Academy of Sciences, 118(29), e2015367118. https://doi.org/10.1073/pnas.2015367118

Sunderlin, W. D., Larson, A. M., Duchelle, A. E., Resosudarmo, I. A. P., Huynh, T. B., Awono, A., & Dokken, T. (2014). How are REDD+ Proponents Addressing Tenure Problems? Evidence from Brazil, Cameroon, Tanzania, Indonesia, and Vietnam. World Development, 55, 37–52. https://doi.org/10.1016/j.worlddev.2013.01.013

Uisso, A. J., Chirwa, P. W., Ackerman, P. A., & Mbwambo, L. (2021). Non-carbon Benefits as Incentives for Participation in REDD + and the Role of Village Participatory Land Use Plans in Supporting This: Insights from Kilosa District, Tanzania. Journal of Environmental Planning and Management, 64(6), 1111–1132.

Wright, G. D., Andersson, K. P., Gibson, C. C., & Evans, T. P. (2016a). Decentralization can help reduce deforestation when user groups engage with local government. Proceedings of the National Academy of Sciences, 113(52), 14958–14963.

Minor:

1. Page 4 (line129). R Studio should be replaced with R (the underlying language used in R Studio). Moreover, the authors should cite the packages used in their R analyzes. 

2. Standards are typically understood as command-and-control if a technology or quota are imposed. Consider this as you use standards in your paper.  Is there a way of combining standards, certifications, and eco-labelling in one term in section 3.2.4? What are the differences between the 3 terms?

3. In section 3.2.6, you may want to speak about ESG.

Author Response

Response to Reviewer 1 Comments

Point 1: The forest policies described in the paper are not all market-based instruments. In environmental economics, the MBI include PES, taxes, subsidies, and cap-and-trade programs (see environmental economics textbooks). Some of policies listed in this review do not have a market component (e.g., education campaigns, logging bans). The paper should change MBIs to environmental policies (or forest policies) or at a minimum explain why the other mechanisms are considered MBI. In section 3.2.10, the paper acknowledges that CC and IE are not part of MBIs, but they are part of your review. Thus, you may consider changing the focus of the paper and title to be based on forest policies.

Response 1: We completely agree with the reviewer that CC and IE are not part of MBIs. Therefore, we decided to remove those policy instruments which do not belong to MBI from the manuscript, rather than completely shifting the focus to forest policies and changing the title of the manuscript.

Point 2: What's the difference between a PES and offsets? Isn't there an overlap? There could be a corporation, NGO, or government paying for an ecosystem service as an offset.

Response 2: Good Point! PES is almost similar to offset schemes except offset schemes are either driven by regulatory caps in the emission trading scheme or voluntary caps in the voluntary carbon market. Yes, there is an overlap between PES and offset schemes and the boundary between some MBIs is blur. This is why we decided to apply the theory of policy mix in the discussion part. We have clarified this point in the manuscript. Please see lines 287-289.

Point 3: Actors (section 3.3). It would be better to find a more descriptive name for the third sector (e.g. non-profits) that is analogous to the other sectors.

Response 3: Thank you for the suggestion. The “third sector” has been replaced by the label “civil society”. Non-profit has already been used as distinguishing criteria, so the label “civil society” is used instead. See lines 350-351 and Figure 4.

Point 4: Figure 4 is hard to read. Is the circle of the third sector supposed to be contained within the triangle such that the circle is within formal and non-profit lines?

Response 4: Good catch. Figure 4 has been modified to make it readable. Third sector in our case include entities such as social entrepreneurship, non-profit organizations, cooperative organizations, which can be viewed as cornerstone for state, market, community sector. We argue that it is more than formal and non-profit organizations.

Point 5: The paper will benefit from some discussion about land tenure structures in the meta-analysis. Are there formal/informal? Are there communal land tenure structures in the area? The latter analysis will enhance the metanalysis.

Response 5: Thank you for pointing out missing discussion about land tenure structures. We have to say that not all the literature reviewed in this study have information about land tenure structures since this requires more in-depth analysis of projects. However, we have given some examples of forest carbon mechanisms having tenure insecurity problems in the discussion part. Please see lines 505-517.

Point 6: What are measures of goodness of fit within the clusters? In Figure 5, cluster 1 have closer points relative to clusters 2 and 3. Comments about the cluster quality should be included

Response 6: This is done. We have added a bar chart of inertia gain in Figure 5 to give more illustration. Inertia measures how well a dataset is clustered by K means. Detailed description is shown in lines 424-426 and 438-441.

Point 7: Limitations: within the section of future research, limitations of this article should be specified.

- For example, does focusing on articles written in English result in a sample selection that does not offer a wider representation of research in the global south.

- This paper doesn't include reports from key organizations such as the CIFOR (https://www.cifor.org/). This should be acknowledged at some point.

Response 7: Thank you for your insightful comments. The above limitations have been mentioned in the manuscript. Please see lines 697-700.

Point 8: The following peer-reviewed articles from top environmental and natural resource economics journals are missing from this review:

Response 8: Thank you for additional literature and these are very useful. However, I have to say that we did not include those in our study. The aforementioned literatures were not found using the defined systematic literature review approach. Thus, in order to keep our research consistent and solid, we decided not to include these references in our study. However, we have to say that “the mechanisms reviewed in our study are not an exhaustive list of mechanisms using MBI for forest carbon” and we have already acknowledged this limitation in the Future Research section. But still thank you so much for your insightful comments and some of the literature has been carefully reviewed and used in the discussion.

Minor revision:

Point 1: Page 4 (line129). R Studio should be replaced with R (the underlying language used in R Studio). Moreover, the authors should cite the packages used in their R analyzes.

Response 1: We completely agree with the reviewer that the citation of the R package should be included in the manuscript. This has been addressed and R package—FactoMineR has been cited in the manuscript. Please see reference #36.

Point 2: Standards are typically understood as command-and-control if a technology or quota are imposed. Consider this as you use standards in your paper.  Is there a way of combining standards, certifications, and eco-labelling in one term in section 3.2.4? What are the differences between the 3 terms?

Response 2: We would like to thank the reviewer for pointing out that we should clarify among these terms. We understand three concepts through supply chain. Standards can be viewed as guidelines or criteria that needs to follow. Certification is a procedure by which third-party organizations give written assurance to either process or products. Eco-labelling is a symbol showing the compliance with standards has been met, which provides an easy way for end-consumers to understand complicated standards. We have specified the difference between the 3 terms in the article. Please see lines 266-272.

Point 3: In section 3.2.6, you may want to speak about ESG.

Response 3: Following your suggestion, the concept of ESG has been now introduced and mentioned in section 3.2.6. We defined this term and introduced environmental factors that this concept taken into account. Please see lines 304-308.

Reviewer 2 Report

The authors achieved their own declared aim of the paper to a very large degree. The aim of the paper was to map global and up-to-date empirical patterns of MBIs, quantitatively identify and discuss policy design features and unravelling the role of institutions in the governance of MBIs. Overall, the work is well documented and presented at high scientific level. The topic of the paper is highly policy relevant as MBI for climate change mitigation are an emerging phenomenon but also among the most controversially discussed instruments, especially when addressing forest ecosystems. The authors did not address challenges related to forest carbon markets involving forest ecosystems, what can be justified by the specific focus on the role of public institutions.

However, I am concerned that the paper is not sufficiently shaping its conclusions towards answering policy relevant questions. The authors claim in the conclusions that “These insights could be useful for policymakers to understand their responsibility in climate policy.” However, the study does not address in the interpretation of results the current needs of policy makers in the field. In the first line of the conclusions, the authors state that “Forest carbon plays an important role in mitigating climate change.” However, the analysis completely ignores current frameworks being set for MBIs for climate change mitigation.

The authors do not refer a single time to current policy developments that have the potential to be a game changer for such markets. This is, e.g. CORSIA and Art. 6 of the Paris Agreement. There two processes form future arenas where public institutions need to play an increasing role. This due to the close linkages that mitigation projects in the future will have to have with national targets and national governance. This is why so called “nested” (ensuring consistency between projects and higher-level targets) and “jurisdictional” approaches (putting not projects but larger administrative units into focus) are discussed.

Due to the missing linkage to these policy developments, the conclusions drawn appear to be one-dimensional and detached from current policy context. I strongly recommend including the current policy processes mentioned in introduction, analysis and conclusions to make sure, the results contribute to currently ongoing debates where they are very much needed.

Minor issues detected:

line 29             forest ecosystems

line 94             Lite-rature

line 352           consider revision of incomplete phrase: “Market mainly consists of private sector.”

Figure 4          Too small font, hardly readable text in oval shapes

Author Response

Response to Reviewer 2 Comments

Point 1: The authors achieved their own declared aim of the paper to a very large degree. The aim of the paper was to map global and up-to-date empirical patterns of MBIs, quantitatively identify and discuss policy design features and unravelling the role of institutions in the governance of MBIs. Overall, the work is well documented and presented at high scientific level. The topic of the paper is highly policy relevant as MBI for climate change mitigation are an emerging phenomenon but also among the most controversially discussed instruments, especially when addressing forest ecosystems. The authors did not address challenges related to forest carbon markets involving forest ecosystems, what can be justified by the specific focus on the role of public institutions.

Response 1: We would like to thank the anonymous reviewer for his/her insightful comments. We really appreciate his/her effort in reviewing our manuscript.

Point 2: However, I am concerned that the paper is not sufficiently shaping its conclusions towards answering policy relevant questions. The authors claim in the conclusions that “These insights could be useful for policymakers to understand their responsibility in climate policy.” However, the study does not address in the interpretation of results the current needs of policy makers in the field. In the first line of the conclusions, the authors state that “Forest carbon plays an important role in mitigating climate change.” However, the analysis completely ignores current frameworks being set for MBIs for climate change mitigation.

Response 2: We completely agree with reviewer that we should better shape our conclusion towards answering policy questions. Therefore, we have decided to add multiple lines about the implications for policy-makers in the conclusion section. Please see lines 722-734. 

Point 3: The authors do not refer a single time to current policy developments that have the potential to be a game changer for such markets. This is, e.g. CORSIA and Art. 6 of the Paris Agreement. There two processes form future arenas where public institutions need to play an increasing role. This due to the close linkages that mitigation projects in the future will have to have with national targets and national governance. This is why so called “nested” (ensuring consistency between projects and higher-level targets) and “jurisdictional” approaches (putting not projects but larger administrative units into focus) are discussed.

Response 3: Thank you for your insightful comments. Following this suggestion, we have added a new paragraph about the increasing and changing role of public institutions in the future policy arena, for the cases of the Art. 6 of the Paris Agreement and the future inclusion of forest carbon offsets in the EU ETS. Please see lines 672-692. We have also mentioned the close linkage between country’s NDC and mitigation projects and forest carbon markets become more and more institutionalized.

Point 4: Due to the missing linkage to these policy developments, the conclusions drawn appear to be one-dimensional and detached from current policy context. I strongly recommend including the current policy processes mentioned in introduction, analysis and conclusions to make sure, the results contribute to currently ongoing debates where they are very much needed.

Response 4: We would like to thank reviewer for pointing out the missing policy context in the manuscript. We have now mentioned the importance of policy context in the introduction and the notion of why we need to mention policy development. We have also given a few examples of forest carbon mechanisms operating under policy frameworks. Please see lines 39-47. In the discussion, we added in the text how the finalization of the Art. 6 of Paris agreement shape the role of public institutions and what role public authorities will play in the future. We have also mentioned the inclusion of forest investments in EU ETS will make forest carbon market more institutionalized. Please see 672-692. In the conclusion, we have emphasized the increasing role of public institutions in the future climate policy arena. Please see lines 727-734.
